# Active Layer Thickness Estimation from X-Band SAR Backscatter Intensity

Barbara Widhalm[1], Annett Bartsch[1,2], Marina Leibman[3,4], and Artem Khomutov[3,4]

[1]Zentralanstalt für Meteorologie und Geodynamik, 1190 Vienna, Austria
[2]Vienna University of Technology, 1040 Vienna, Austria
[3]Earth Cryosphere Institute, Russian Academy of Sciences, Siberian branch, Tyumen 625000, Russia
[4]Tyumen State University, Tyumen 625006, Russia

*Correspondence to:* Barbara Widhalm (barbara.widhalm@zamg.ac.at)

**Abstract.** The active layer above the permafrost, which seasonally thaws during summer is an important parameter for monitoring the state of permafrost. Its thickness is typically measured locally, but a range of methods, which utilize information from satellite data, exist. Mostly, the Normalized Difference Vegetation Index (NDVI) obtained from optical satellite data is used as proxy. The applicability has been demonstrated mostly for shallow depths of Active Layer Thickness (ALT) below approximately 70 cm. Some permafrost areas including central Yamal are, however, characterized by larger ALT. Surface properties including vegetation structure are also represented by microwave backscatter intensity. So far, the potential of such data for estimating ALT has not been explored. We therefore investigated the relationship between ALT and X-Band SAR backscatter of TerraSAR-X (averages for 10 x 10 m window) in order to examine the possibility of delineating ALT on a continuous and larger spatial coverage in this area and compare it to the already established method of using NDVI from Landsat (30 m). Our results show that the mutual dependency of ALT and TerraSAR-X backscatter on land cover types suggests a connection of both parameters. A range of 5 dB can be observed for an ALT range of 100 cm (40 - 140 cm) and an R² of 0.66 has been determined over the calibration sites. An increase of ALT with increasing backscatter can be determined. The RMSE over a comparably heterogeneous validation site with maximum ALT of > 150 cm is 20 cm. Deviations are larger for measurement locations with mixed vegetation types (especially partial coverage by cryptogam crust) with respect to the spatial resolution of the satellite data.

## 1 Introduction

Permafrost is defined as soil or rock that remains at or below 0°C for two or more consecutive years (Harris et al., 1988) and currently underlies some 25 % of the Earth's land surface (Huggett, 2007). Due to global warming extensive areas where permafrost is presently within a degree or two below the melting point could be destabilized (Smith, 1990). At global scale, increased ground temperatures could facilitate further climatic changes by releasing greenhouse gases that are currently sequestered in the upper layer of permafrost by increasing the annual thaw depth (Kane et al., 1991; Gomersall and Hinkel, 2001; Shiklomanov and Nelson, 1999; Schaefer et al., 2011; Schuur et al., 2015). The top layer of ground subject to annual thawing and freezing in areas underlain by permafrost is defined as the active layer (Permafrost Subcommittee, 1988). In this layer

most ecological, hydrological and biochemical activities take place (Kane et al., 1991; Brown et al., 2000). Furthermore it is an essential climate variable to monitor permafrost regions (Schaefer et al., 2015), making it not only an important factor at regional but also global scale. The active layer thickness (ALT) is predominately controlled by ambient temperature, but is also influenced by insulation layers such as snow cover and vegetation, slope, drainage, soil type, organic layer thickness and water

content (Leibman, 1998; Shiklomanov and Nelson, 1999; Hinkel and Nelson, 2003; Kelley et al., 2004; Melnikov et al., 2004; Vasiliev et al., 2008). Due to the interaction between these surface and subsurface factors that can be spatially highly variable, ALT may vary substantially over short lateral distances (Shiklomanov and Nelson, 1999; Leibman et al., 2012).

Near-surface permafrost area is projected to decrease within the next century (IPCC, 2013). Changes in active layer thickness have been already observed for Yamal (Leibman et al., 2015), where active layer thickness spatial patterns are unknown outside

of the sites with in situ measurements.

Analytical procedures exist to estimate ALT, such as the Stefan solution (Harlan and Nixon, 1978) or the Kudryavtsev equation. While the Stefan solution links the seasonal thaw depth to the accumulated surface thawing-degree days, the Kudryavtsev equation accounts for the effects of snow cover, vegetation, soil moisture, thermal properties, and regional climate (Kudryavtsev et al., 1974; Yershov, 1998; Shiklomanov and Nelson, 1999). These methods, although accurate, are labour intensive and

limited in spatial coverage (Gangodagamage et al., 2014).

While traditional in situ measuring methods like probing with metal rods are very inefficient at regional scale, remote sensing holds great potential to delineate ALT on a continuous and larger spatial coverage. ALT can be derived by empirical relationships between probe measurements and a physical attribute measurable by remote sensing (Schaefer et al., 2015). Investigations have been made using the Normalized Difference Vegetation Index (NDVI) (McMichael et al., 1997; Kelley

et al., 2004), digital elevation data and land-cover classes (Nelson et al., 1997; Peddle and Franklin, 1993). Especially optical data have been used to retrieve vegetation characteristics (see Table 1). A combination with derivatives of digital elevation models has been shown to be of added value (Peddle and Franklin, 1993; Leverington and Duguay, 1996; Gangodagamage et al., 2014). Application of high resolution optical satellite data in combination with high resolution digital elevation data from airborne measurements have been shown to be useful for mapping ALT but are very limited in spatial extent. Subsurface

thermal properties are also derived from landcover classes, but applicability has been demonstrated only for the permafrost transition zones so far (**?**).

Recently, subsidence rates have been used as input for modelling ALT (Schaefer et al., 2015). Synthetic Aperture Radar has been exploited using interferometric analyses (InSAR, provides seasonal ground subsidence) in combination with soil properties to estimate ALT without using empirical relationships with probing data (Schaefer et al., 2015).

Most previous remote sensing approaches (Leverington and Duguay, 1996; McMichael et al., 1997; Sazonova and Romanovsky, 2003; Schaefer et al., 2015) have utilized data with spatial resolutions of 30 m and coarser and were conducted in areas with shallow ALT (less than approximately 70 cm, see Table 1). Many regions such as the Yamal peninsula are however characterized by a larger ALT range. Deeper active layers were modeled by analytical approaches (e.g., Sazonova and Romanovsky, 2003) or by incorporating only a few ALT classes (e.g., Leverington and Duguay, 1996). For Yamal, previous tests

indicate that ALT below 70 cm, 70 - 100 cm and above 100 cm can be distinguished using NDVI (Leibman et al., 2015).

SAR backscatter intensity has so far not been investigated for ALT estimation. Radar backscatter at X-band is also related to vegetation coverage, especially shrubs (Duguay et al., 2015), due to volume scattering, similarly to the NDVI. The microwave signal interacts with leafs in the shrub canopy due to the comparable short wavelength ($\sim 3$ cm). Leaf size of willows often exceed this wavelength and stems can have significant larger diameters (Widhalm et al., 2016). Also vegetation water content may influence the return signal, but no dedicated investigations for tundra species exist. The overall backscatter intensity of a certain surface area is also influenced by surface roughness with respect to the wavelength, as well as by soil moisture. A higher moisture content leads to a higher dielectric constant and therefore higher backscatter values and additionally a reduction in penetration depth. X-band investigations in tundra (Lena Delta) suggest that such variations can occur (**?**), but results from **?** showed no sensitivity to precipitation over the same area.

In this study we hypothesize that there is a relationship between local ALT and X-band measurements, based on the influence of ALT affecting features on backscatter strength. Like ALT, backscatter is dependent on vegetation cover. Furthermore shrubs, which can be monitored with X-band data, can be related to snow cover due to their characteristic of retaining snow (Domine et al., 2016). Moreover the terrain in this region is closely linked to drainage and water content which are influencing ALT. These characteristics are in turn influencing the predominant vegetation, which, including soil moisture, can be captured with microwave backscatter. We use in situ records from several sites in the proximity of a long-term monitoring site on Yamal and discuss the results with respect to previous approaches which use remotely sensed information as proxy for ALT.

## 2 Study area and datasets

### 2.1 The Vaskiny Dachi monitoring site

The Vaskiny Dachi research station (70°20'N, 68°51'E) was established in 1988 and is situated in the central Yamal Peninsula in a system of highly-dissected alluvial-lacustrine-marine plains and terraces. It is located within a region of continuous permafrost where tundra lakes and river flood plains are the most prominent landscape features (Leibman et al., 2015). Dense dwarf shrubs (Betula nana) are widespread on the watersheds. Well-drained hilltops are occupied by dwarf shrub-moss-lichen communities. On gentle poorly drained slopes, low shrubs and dwarf shrubs are well developed and mosses predominate. On convex tops and windy hill slopes, shrub-moss-lichen communities with spot-medallions are predominant. River valleys, thermocirques, and landslide cirques with thick snow cover are characterized by willow thickets. Sedge and sphagnum bogs and flat-topped polygonal peatlands are common on flat and concave (saddles) surfaces of watersheds and terraces, in the river valley bottoms, on low lake terraces and in other depressions (Khomutov and Leibman, 2014).

The study area is characterized by continuous permafrost. ALT ranges between 40 cm in peat and up to 120 cm on sandy, poorly vegetated surfaces (Melnikov et al., 2004; Vasiliev et al., 2008; Leibman et al., 2011, 2012). There are extremes observed on high-center sandy polygons, which can be 1–1.5 m high and up to 10 m in diameter, with active layer exceeding 2 m. Spatial changes in ground temperature are controlled by the redistribution of snow which results from strong winds characteristic for tundra environments and the highly dissected relief of Central Yamal (Dvornikov et al., 2015). Lowest ground temperature is characteristic for hilltops with sparse vegetation where snow is blown away. The warmest are areas with high willow shrubs,

**Table 1.** Studies which used satellite data for determination of active layer thickness (ALT): type of satellite data, accuracy and active layer ranges

| Reference | remotely sensed data | ALT measurement | resolution | study area | ALT | accuracy |
|---|---|---|---|---|---|---|
| Peddle and Franklin (1993) | SPOT imagery (land cover) and photogrammetric DEM | in situ soil probing at regular intervals throughout field sample sites of 60 x 60 m | 20 m | Ruby Range, southwest Yukon Territory, Canada | classes: < 25 cm; 25 cm - 50 cm; > 50 cm; no permafrost | 79% agreement for the four classes |
| Leverington and Duguay (1996) | Landsat Thematic Mapper (NDVI, land cover) and DEM data | pit measurements supplemented by three ground probing measurements in 5 m intervals in four directions at each site | 30 m | Mayo region, Central Yukon Basin, Canada | classes: < 70 cm; 70 cm - 150 cm; > 150 cm | 93% agreement for the three ALT classes |
| McMichael et al. (1997) | Landsat Thematic Mapper and handheld radiometer (NDVI) | median of five ground probing measurements at each sample point | 30 m | Prudhoe Bay and Happy Valley, North Slope, Alaska | average: < 35 cm | No relationship between NDVI and ALT on the North Slope of Alaska in areas with little variation in relief. In areas where topography strongly controls the flow and redistribution of water, NDVI did account for approximately 40% of the variability |
| Gangodagamage et al. (2014) | LiDAR (local slope and landscape curvature) and Worldview-2 (NDVI) | probing along transects of diverse length and at various intervals | 2 m | Barrow, Alaska | 20 - 70 cm | $R^2 = 0.76$ and RMSE ± 4.4 cm |
| Schaefer et al. (2015) | Subsidence from multiannual Phased Array type L-band Synthetic Aperture Radar (PALSAR) | average of calibrated Ground Penetrating Radar (GPR) measurements within each pixel (∼ 40 traces per pixel); probe measurements at two CALM sites (1 x 1 km grid with 100 m interval, 10 x 10 m plot random point placement) | ∼ 30 m | Barrow, Alaska | average: 30 - 40 cm; < 40 cm in areas outside of drained lake basins | ∼ 76% of the study area within uncertainty of the used Ground Penetrating Radar and probing data (∼ 8 cm) |

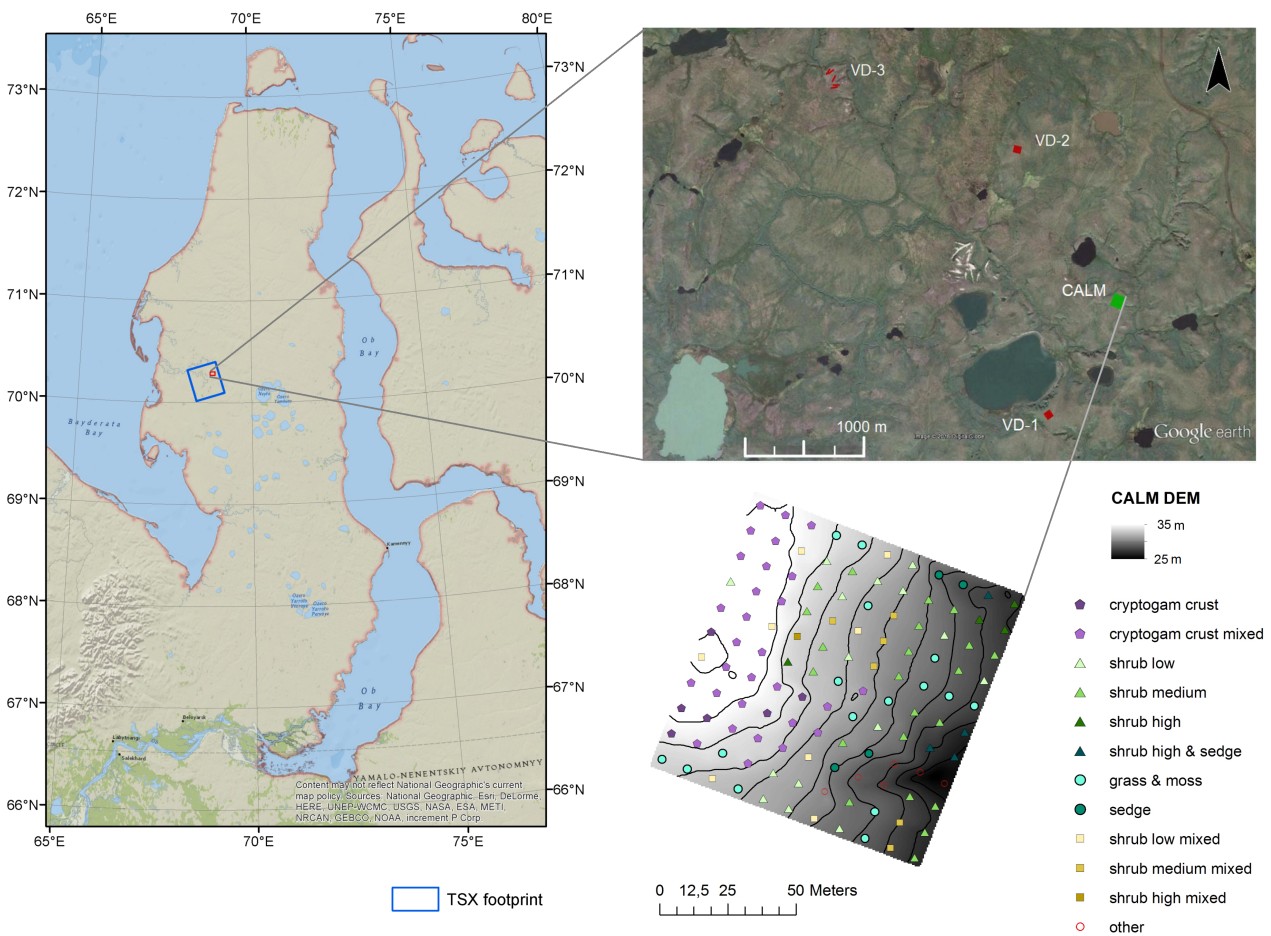

**Figure 1.** Location of the study area (with monitoring sites Vaskiny Dachi (VD) 1-3 and CALM gird) within the Yamal peninsula and CALM grid DEM with land cover information (Sources left: ArcMap Basemap: National Geographic, Esri, DeLorme, HERE, UNEP-WCMC, USGS, NASA, ESA, METI, NRCAN, GEBCO, NOAA, increment P Corp. Source top right: Google Earth, Image © 2016 Digital Globe. Bottom right: DEM Dvornikov et al. (2016),land cover information from vegetation survey of August 2015)

due to the retention of snow, found on slopes, in valleys and lake depressions. While the spatial distribution of ALT depends on lithology and surface covers, temporal fluctuations are controlled by ground temperature, summer air temperature and summer precipitation (Leibman et al., 2015).

## 2.2 In situ measurements

In 1993 a Circumpolar Active Layer Monitoring (CALM) site was established at Vaskiny Dachy, placed on the top and slope of a highly dissected plain, affected by landslides, with sandy to clayey soils. The CALM program, designed to observe the response of the active layer and near-surface permafrost to climate change, currently incorporates more than 100 sites. The

International Permafrost Association serves as the international facilitator for the CALM network, which is now part of the WMO Global Terrestrial Network for Permafrost (GTN-P) (Brown et al., 2000).

Within the Greening of the Arctic (GOA) project of the International Polar Year (IPY), which was funded by NASA's Land-Cover Land-Use Change (LCLUC) program, three additional monitoring sites were established in the Vaskiny Dachi area (Walker et al., 2009). Study sites were established within areas with more or less homogeneous vegetation. The site Vaskiny Dachi-1 (VD-1) has clay soils and the vegetation is heavily grazed sedge, dwarf-shrub-moss tundra. Soils at Vaskiny Dachi-2 (VD-2) are a mix of sand and clay, its vegetation is heterogeneous, but dominated by dwarf birch, small reed grass and sedge, cowberries and mosses. At Vaskiny Dachi-3 (VD-3) the soils are sandy and the vegetation is a dry dwarf-shrub-lichen tundra (Walker et al., 2009).

The ALT is measured by a metal probe according to CALM protocol. This involves a late season mechanical probing, in this case late August, when ALT is near its end-of-season maximum. A 1 cm diameter graduated steel rod is inserted into the soil to the depth of resistance to determine the depth of thaw (Brown et al., 2000).

ALT is measured at a spacing of 10 m within the 100 x 100 m grid at the CALM site, resulting in 121 measuring points. The VD sites feature 5 transects respectively. At VD-1 and VD-2 these transects form grids of 50 x 50 m. Transects are 12.5 m apart and ALT is measured every 5 m, resulting in 55 measurement points per site. The transects at VD-3 are arranged to areas of homogeneous vegetation (Walker et al., 2009). The site of VD-3 features higher ALT values, most likely because of the present sandy soils, which yield a greater conductivity and water permeability (higher convective heat exchange). The CALM grid site is far more heterogeneous than the other VD sites and holds patches of dry cryptogam crust, grasses and mosses, low and high shrubs as well as some wet sedge spots. Cryptogam crust is encountered at the concave hilltop, while high shrubs were mostly located at the landslides (Figure 8). Here, ALT is locally higher due to high salinity of clayey deposits which contain no ice under negative temperature and do not resist to probing.

In August 2015, we carried out a dedicated vegetation survey of each CALM grid point, where we determined the dominant vegetation cover within a 3 x 3 m area (Figure 1). The following classes are distinguished: Cryptogam crust, low shrubs (< 15 cm), medium shrubs (15 -30 cm), high shrubs (> 30 cm), grass and moss as well as a class where sedges dominate and classes of mixed vegetation. Further information on vegetation has been collected outside of the ALT measurement sites in August 2014. Over 60 points were registered but only 36 of these points could be further used because of low homogeneity with respect to the spatial resolution of the satellite data. This survey included the most dominant classes of the region and therefore covered more types than can be found at the VD sites or the CALM grid: low shrubs (< 20 cm), medium shrubs (20 -60 cm), high shrubs (> 60 cm), Cryptogam crust, and a mixture of grasses and sedges.

Furthermore we conducted moisture measurements at the CALM grid. We used the Delta-T Wet Sensor with HH2 handheld to measure the moisture content of the top 5 cm at each grid point on three dates in August 2015.

## 2.3 X-band data

X-band data has so far been used to investigate surface deformations like thaw subsidence and frost heave (Beck et al., 2015; Zhang et al., 2016; Wang et al., 2016), for monitoring tundra shrub growth (Duguay et al., 2015) or dating drained thermokarst lake basins (Regmi et al., 2012) related to permafrost research.

Backscatter information from data of the TerraSAR-X mission have been used for this study. The digital elevation model available from the TanDEM-X mission has been used as auxiliary data set for TerraSAR-X data preprocessing.

The German national SAR-satellite system TerraSAR-X is based on a public-private-partnership agreement between the German Aerospace center DLR and EADS Astrium GmbH. It was launched in June 2007 and started its operational service at the beginning of 2008 (DLR, 2009). The satellite flies in a sun-synchronous, dawn dusk orbit with an 11-day repeat period.

TerraSAR-X features an advanced high-resolution X-Band Synthetic Aperture Radar with a centre frequency of 9.65 GHz corresponding to a wavelength of about 3.1 cm. TerraSAR-X operates in Spotlight-, Stripmap- and ScanSAR Mode with various polarizations. In this study HH (horizontally sent and horizontally received) polarized images of Stripmap mode were used, which image strips of 30 km width at 3 m resolution (Werninghaus et al., 2004).

Six images from August 2014 and 2015 (three images per year) were obtained as SSC (Single Look Slant Range Complex)

and have been acquired in the same ascending orbit and beam (incidence angle range 27.3° - 30.3°).

The TanDEM-X mission is an extension of the TerraSAR-X mission, coflying a second satellite of nearly identical capability in a close formation. This enables the acquisition of highly accurate cross- and along-track interferograms without the inherent accuracy limitations imposed by repeat-pass interferometry due to temporal decorrelation and atmospheric disturbances (Krieger et al., 2007). In this study the TanDEM-X Intermediate DEM (IDEM, $\sim$ 12 m pixel spacing, < 10 m absolute hori-

zontal and vertical accuracy) was used for terrain correction, which, compared to the final TanDEM-X DEM product, might have limitations with respect to product quality and completeness (DLR).

## 2.4 Landsat data

Landsat 8, launched in 2013, is a NASA (National Aeronautics and Space Administration) and USGS (Department of the Interior U.S. Geological Survey) collaboration, which extends the 40 year Landsat record. It carries two sensors, the Operational

Land Imager (OLI) and the Thermal Infrared Sensor (TIRS), which spectral bands remain comparable to the Landsat 7 ETM+ and operate in the visible, near-infrared, short wave and thermal infrared. Landsat 8 flies in a near-polar, sun-synchronous 705 km circular orbit and acquires data in 185 km swaths segmented into 185 km x 180 km scenes. For this study two Level 1 terrain-corrected (L1T) scenes of $22^{nd}$ July 2014 and $10^{th}$ August 2015 were obtained in order to calculate NDVI (spatial resolution 30 m) and compare the already established approach of using NDVI for ALT delineation to the in this study

introduced approach of utilizing TerraSAR-X backscatter.

## 3 Methodology

In Microwave Remote Sensing the radar backscatter is dependent on sensor parameters like incidence angle, polarization and wavelength and also on geometric parameters such as surface roughness and vegetation structure, as well as soil properties. Shorter wavelengths do not penetrate as much as longer wavelengths into vegetation and soil, therefore short wavelengths like X-band rather yield information about the upper layers of vegetation (Ulaby et al., 1982). The assumption for this study is that surface roughness variations play a minor role regarding spatial backscatter differences across the study site. Backscatter increases with increasing vegetation height for X-band in tundra what originates from volume scattering and double bounce (leading to higher backscatter) rather than surface roughness (Ullmann et al., 2014). It can be also expected that soil moisture variations are not reflected in X-band measurements when vegetation cover is present. The assumption is that volume scattering and double bounce in vegetation is the main contributor to spatial differences in backscatter. Several studies exemplified the separability of tundra shrubs from their surroundings (Ullmann et al., 2014; Duguay et al., 2015). In **?** it is shown that X-band time series analysis allows a clear discrimination of major landscape elements in tundra regions.

The local vegetation patterns are influenced by terrain and soil moisture and also correlate with snow cover thickness, which are all ALT influencing factors (Shiklomanov and Nelson, 1999; Gomersall and Hinkel, 2001; Kelley et al., 2004). Areas with shrubs have higher snow cover, which prevents the ground from cooling in the winter. The ALT might be therefore also higher than in the surrounding area. Based on results of previous studies which utilized the vegetation index NDVI (Table 1) and the mutual dependency of NDVI and radar backscatter on vegetation coverage, it is expected that a relationship between X-Band backscatter measurements and ALT is given. We compiled a data set based on TerraSAR-X which allows the investigation of this relationship. This included for SAR common pre-processing steps to account for variations due to viewing geometry.

Six TerraSAR-X images from August 2014 and 2015 were processed (three images per year). It is assumed that within this time stable phenological conditions can be expected. Utilising the software NEST (Next ESA SAR Toolbox) Range-Doppler Terrain Correction was performed with a TanDEM-X Intermediate DEM ($\sim$ 12 m resolution) in order to orthorectify the images and to compensate for distortions due to topographical variations and the tilt of the sensor. Images were processed to a pixel spacing of 2 m and a radiometric normalization was applied to account for incidence angle dependent sensitivity. The so called resulting $\sigma_0$ values were then converted into dB. The term backscatter refers in the following to these values, which represent the normalised measure of the radar return. SAR data are affected by so called speckle which is a noiselike effect. It can be understood as an interference phenomenon due to a number of scatterers within each resolution cell. The images were therefore further averaged over time and a spatial filter (average value of the cells in the neighbourhood was calculated: 5 x 5 cells – 10 m) was applied to subdue this noise.

Backscatter as well as NDVI values for the sites VD-1, VD-2 and VD-3 were extracted and compared to the mean ALT values of 2014 and 2015 of each measuring point. The relationship of backscatter values and ALT was examined by varying representations. Bloxplots for ALT classes and for backscatter classes are shown in Figure 4. ALT values were separated into 10 cm classes and the backscatter classes ranged 1 dB, allowing a representative number of sampling points per class (7 and 13 points minimum per class respectively). $\sigma_0$ values as well as NDVI values were additionally compared directly to ALT by scat-

terplots (Figure 5). Fitted linear functions were applied to characterise the relationship between TerraSAR-X backscatter and NDVI respectively and ALT values of the sites VD-1, VD-2 and VD-3. The coefficient of determination ( $R^2 = 1 - \frac{\sum (y_i - f_i)^2}{\sum (y_i - \bar{y})^2}$, the proportion of the variance in the dependent variable that is predictable from the independent variable) was calculated for the regression lines which involved all VD sites, as well as for regression lines for each VD site separately (Table 2). Validation

has been undertaken using ALT measurements at the CALM grid. RMSE was calculated representing the modelled ALT (linear regression of all VD sites) versus the measured ALT at the CALM site. Moreover $\chi^2$ was determined, which also accounts for observational uncertainties. An observational uncertainty of 4 cm was assumed for the probe measurements which has been investigated by Leibman (1998) for sandy soils (2 - 4 cm) in this region. The modelled and measured ALT values at the CALM grid were further compared by scatterplots depicting the different vegetation types (Figure 7) in order to investigate

their impact.

Backscatter statistics have been also derived for different vegetation classes from locations of the 2014 survey outside of the ALT measurement sites (Figure 2). These locations represent relatively homogeneous sites (with respect to TerraSAR-X spatial resolution). The ALT sites, especially the CALM grid, are comparably heterogeneous and therefor of limited applicability for determination of backscatter dependence on vegetation type. The relation of vegetation type and soil moisture was examined

for the CALM grid using moisture measurements of August 2015 (Figure 3).

## 4   Results

The assumption that backscatter increases with increasing amount of vegetation could be confirmed for the Vaskiny Dachi area (Figure 2). There is a difference of about 2 dB between the median for shrubs less than 20 cm and those larger than 60 cm. $\sigma_0$ values for the grass/sedge class do however exceed these values. Cryptogam crust backscatter is at the same order of magnitude

as shrubs between 20 and 60 cm height. Although the sparse vegetation at these areas consists mostly of lichen and volume scattering within is negligible, these spots often show higher surface roughness or even hummocks which may lead to a rise in backscatter amount. The boxplot of Figure 3 which shows the relationship between soil moisture and vegetation types at the CALM grid reveals that lowest soil moisture is encountered for areas with cryptogam crust, while higher shrubs and especially sedges dominate in areas of high soil moisture.

Class statistics (Figure 4) indicate a relationship between $\sigma_0$ and larger thaw depths. Low backscatter values dominate in areas with low ALT and high backscatter values coincide with high ALT. However, the median $\sigma_0$ for shallow ALT does not decrease with decreasing ALT at the same rate as for deeper ALT. The scatterplots of the filtered $\sigma_0$ and ALT values of the sites VD-1 to VD-3 also indicate this correlation (see Figure 5). It also becomes apparent that the change in slope, that is visible in the boxplot of Figure 4, is caused by different backscatter values of the sites VD-1 and VD-2 (Figure 5). Nevertheless

$\sigma_0$ increases generally with increasing ALT. The overall higher ALT values of VD-3 are reflected by their higher backscatter values. A range of 5 dB can be observed for an ALT range of 100 cm (40 - 140 cm).

A coefficient of determination of 0.66 was obtained for the linear regression of backscatter and ALT values (see Table 2). The mathematical form of the regression line is given in Figure 5. The standard error of the intercept is 14.00 and 0.96 for

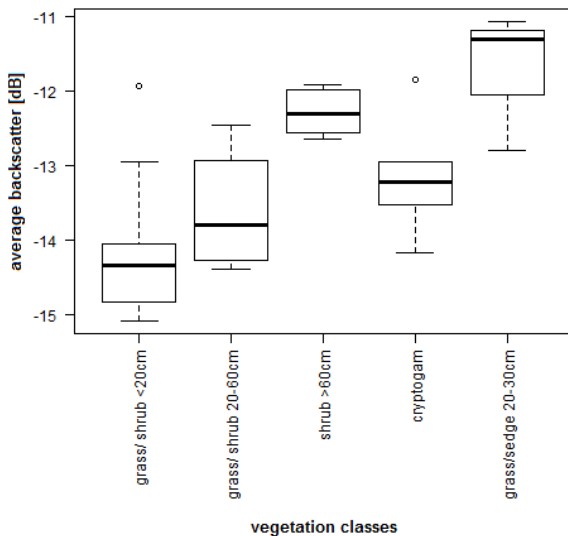

**Figure 2.** Boxplot for backscatter values of various vegetation types. The used points were recorded within a field campaign in 2014. Backscatter values were extracted from the temporally averaged and spatially filtered image of August 2014 and 2015 acquisitions.

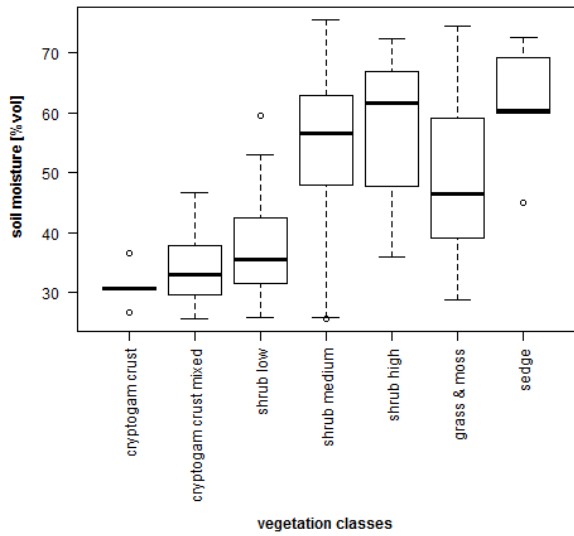

**Figure 3.** Boxplot for mean soil moisture values of three acquisition dates in late August 2015 for vegetation types at the CALM grid.

the slope respectively. $R^2$ for the individual VD sites showed highest values for VD-3 compared to VD-1 and VD-2, which both feature lower ALT values. The linear regression which incorporated all calibration sites (VD-1, VD-2 and VD-3) yielded an RMS error of 20 cm at the CALM grid site and 25 cm for $\chi^2$. Some VD points showed a backscatter variability of more

**Table 2.** $R^2$ between X-band backscatter and NDVI ($22^{nd}$ July 2014 and $10^{th}$ August 2015) respectively and active layer thickness (ALT). $R^2$ has been calculated for all calibration sites combined (VD-1,2 and 3) and separately (ALT range for VD-1: 56 - 108 cm; VD-2: 45 - 102 cm; VD-3: 90 - 140 cm). RMSE values represent the modelled ALT (linear regression of sites VD-1, VD-2 and VD-3) versus the measured ALT at the CALM site. $\chi^2$ statistics compare the modelled and measured ALT at the CALM site under consideration of observational uncertainties.

| dataset | $R^2$ (VD-1, 2 and 3) | RMSE [cm] (CALM) | $\chi^2$ [cm] (CALM) | $R^2$ (VD-1) | $R^2$ (VD-2) | $R^2$ (VD-3) |
|---|---|---|---|---|---|---|
| TSX | 0.656 | **20** | 25 | 0.143 | 0.153 | 0.622 |
| TSX (exclusion of points with high variability) | 0.712 | **21** | 27 | 0.172 | 0.167 | 0.622 |
| NDVI $22^{nd}$ July 2014 | 0.734 | **27** | 46 | 0.171 | 0.016 | 0.017 |
| NDVI $10^{th}$ August 2015 | 0.746 | **30** | 57 | 0.002 | 0.130 | 0.002 |

than 1 dB between the acquisitions of 2014 and 2015. An exclusion of these points would slightly increase the coefficients of determination, but no positive effect could be found with RMSE values at the CALM grid.

The found relationship between TerraSAR-X backscatter and ALT is not that pronounced at the CALM validation site as at the other plots. Especially a patch at the highest elevation of the CALM grid was expected to show higher ALT values according to the calculations from the satellite data (Figure 7 and Figure 6). There are also some spots with slightly higher ALT than predicted. This applies to ALT larger than 125 cm. With the exception of the area around the hilltop the patterns derived with TerraSAR-X however resemble those of the in situ measurements.

A deeper active layer can be found in areas with high shrubs as well as cryptogam crust at the CALM site (see Figure 6). Extremes of ALT can further be encountered at the clay-rich landslides with relatively sparse vegetation, classified as 'other' in Figure 6. Thinner active layers were encountered at zones with grass and moss or low shrubs (see Figure 6). Residuals increase for depths larger than 125 cm, especially in case of dominance of cryptogam crusts.

In Figure 7 a map of the calculated ALT is given for the entire TerraSAR-X scene (open water bodies are masked out). Linear features are expected to be false values as they can be attributed to artificial objects. They show low as well as high ALT values depending on surface roughness and material (air strip and rail track respectively). Values across the scene range between 60 and 180 cm. ALT is higher in drained or partially drained lake basins.

Derived scatterplots of NDVI and ALT values also reveal a relationship within the observed NDVI range of 0.46 – 0.65 (Figure 5). The standard error of the intercept is 9.21 for $22^{nd}$ July 2014 and 11.60 for $10^{th}$ August 2015 and 15.80 and 20.21 for the slope respectively. Both NDVI images showed even larger coefficients of determination of about 0.73 - 0.75, than the TerraSAR-X approach (Table 2). The achieved RMSE at the validation site is however about 7 cm larger than when using the TerraSAR-X backscatter values. Furthermore, when using only site VD-3, which has the highest range and total thickness of the active layer, comparable to the values at the CALM grid, the linear regression for the backscatter values has a coefficient

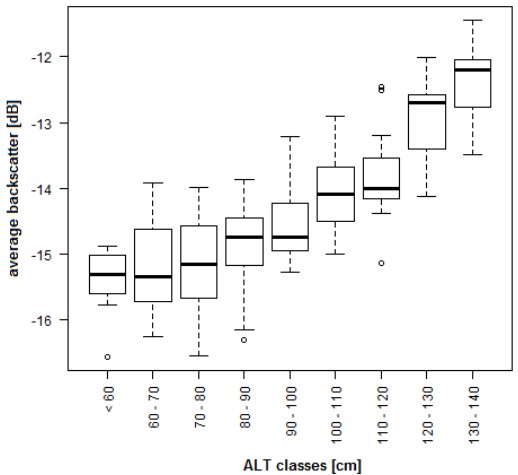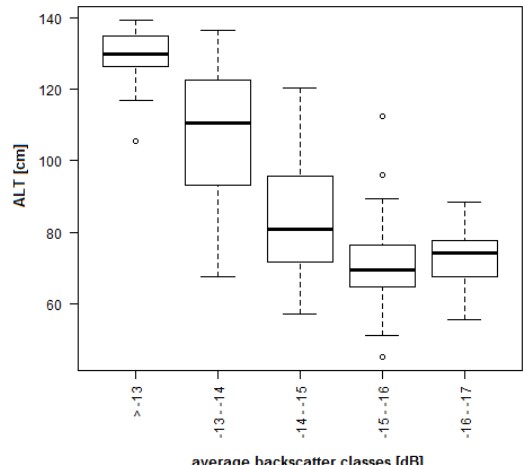

**Figure 4.** Boxplots showing median, minimum and maximum values, first and third quartile and outliers of the ALT and backscatter values of the sites VD-1, VD-2 and VD-3. Left: $\sigma_0$ statistics for Active Layer Thickness (ALT) classes (10 cm). Right: ALT statistics for backscatter classes (1 dB).

of determination of 0.622 while for NDVI values it is only 0.017 and 0.002 respectively (Figure 5). The NDVI derived ALT values for the CALM site range at most between 70 and 100 cm for $22^{nd}$ July 2014 and between 60 and 110 cm for $10^{th}$ August 2015, while measured values have a range of 60 - 150 cm.

## 5 Discussion

5 The assumption that X-band backscatter variations result from mostly differences in volume scattering in vegetation over tundra does not seem to be valid for the selected sites. Radar backscatter is dependent on volume scattering, surface roughness and soil moisture. The influence of all these effects can be perceived in Figure 2. For shrubs a major contributor to backscatter is the volume scattering that takes place within the vegetation. A rise in volume indicated by shrub height clearly gives rise to backscatter amount. This effect is also described in Duguay et al. (2015), where areas with high shrubs show higher backscatter

10 values. A pronounced relationship was however not found for X-band in HH by Ullmann et al. (2014) over the MacKenzie Delta. They applied a smaller spatial filter of 3 x 3 and used data from the end of the growing season (mostly September) when plant decay is already expected to take place. A season related reduction of leaves on shrubs could lead to a decrease in volume scattering and therefore lower backscatter values. The magnitude of backscatter for cryptogam crust, which predominantly occurs on sandy soils, can neither be attributed to volume scattering in vegetation nor to soil moisture (Figure 2 and 3). Surface

15 roughness or partial interaction within the upper soil may lead to higher backscatter than for grass and low shrubs < 20 cm.

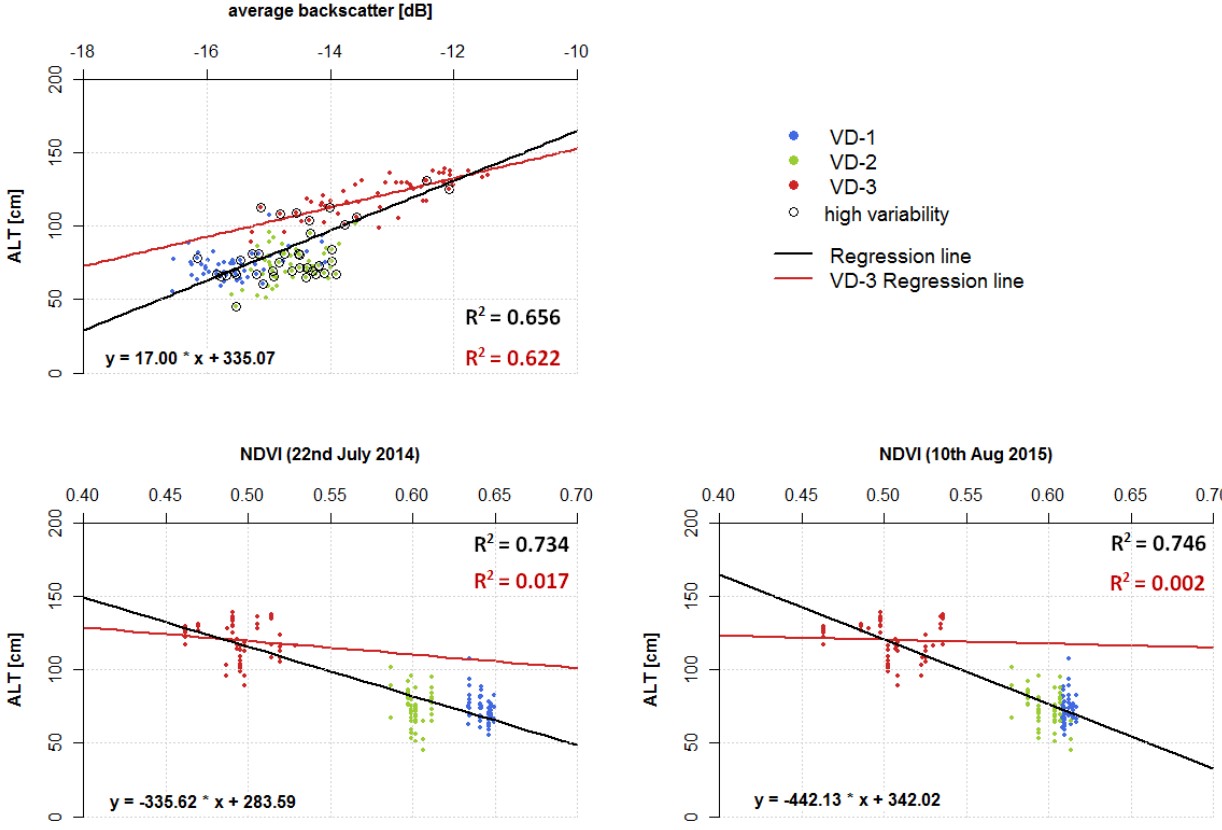

**Figure 5.** Upper Left: Comparison between $\sigma_0$ and Active Layer Thickness (ALT) values at the sites VD-1 to VD-3. Points with differences of more than 1 dB between 2014 and 2015 are marked by circles. Linear Regression for $\sigma_0$ and Active Layer Thickness (ALT) values of all sites in black and for site VD-3 in red. bottom: Comparison between NDVI ($22^{nd}$ July 2014 and $10^{th}$ August 2015) and Active Layer Thickness (ALT) values at the sites VD-1 to VD-3. Linear Regression for NDVI and Active Layer Thickness (ALT) values of all sites in black and for site VD-3 in red.

High soil moisture is typical for the class segdes (Figure 3), giving rise to the backscatter amount. Additionally the so called double bounce effect, which may occur for sedges and grass in standing water, can lead to an increase of received backscatter (Figure 2).

Using backscatter intensity at only one polarization (as available for this study) does not allow to distinguish between surface and volume scattering. Polarimetric analyses (using other combinations of H and V polarizations) may help to distinguish the scattering mechanism contributions (see e.g. Ullmann et al. (2014)). Data acquired at different polarizations are however not available for the study site.

The above described influences on radar backscatter can be linked to ALT controlling factors. The effect of shrubs on ALT has been described contradictorily by different authors. On the one hand shrubs have been said to have a cooling effect due to

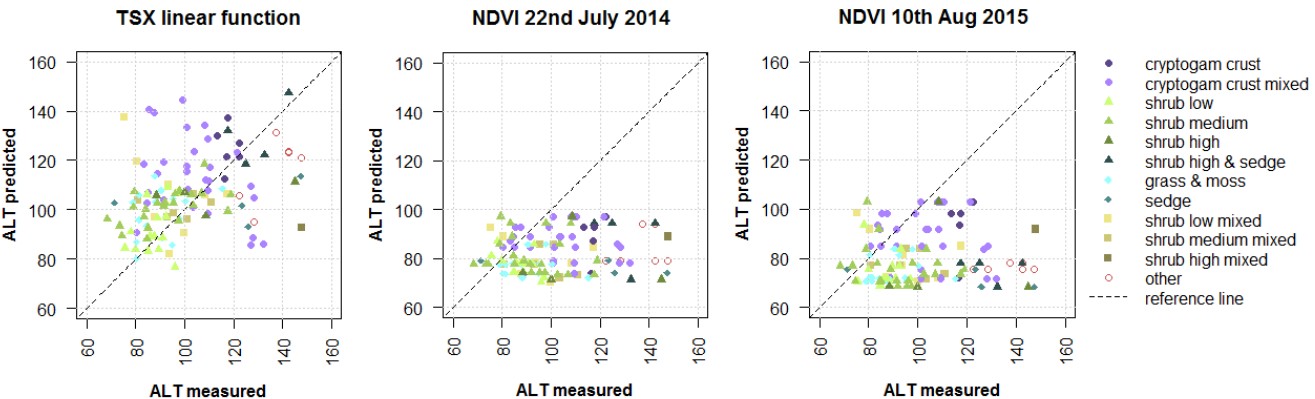

**Figure 6.** Comparison between predicted and measured ALT values at the CALM grid with differentiation between vegetation types for TerraSAR-X and NDVI approaches.

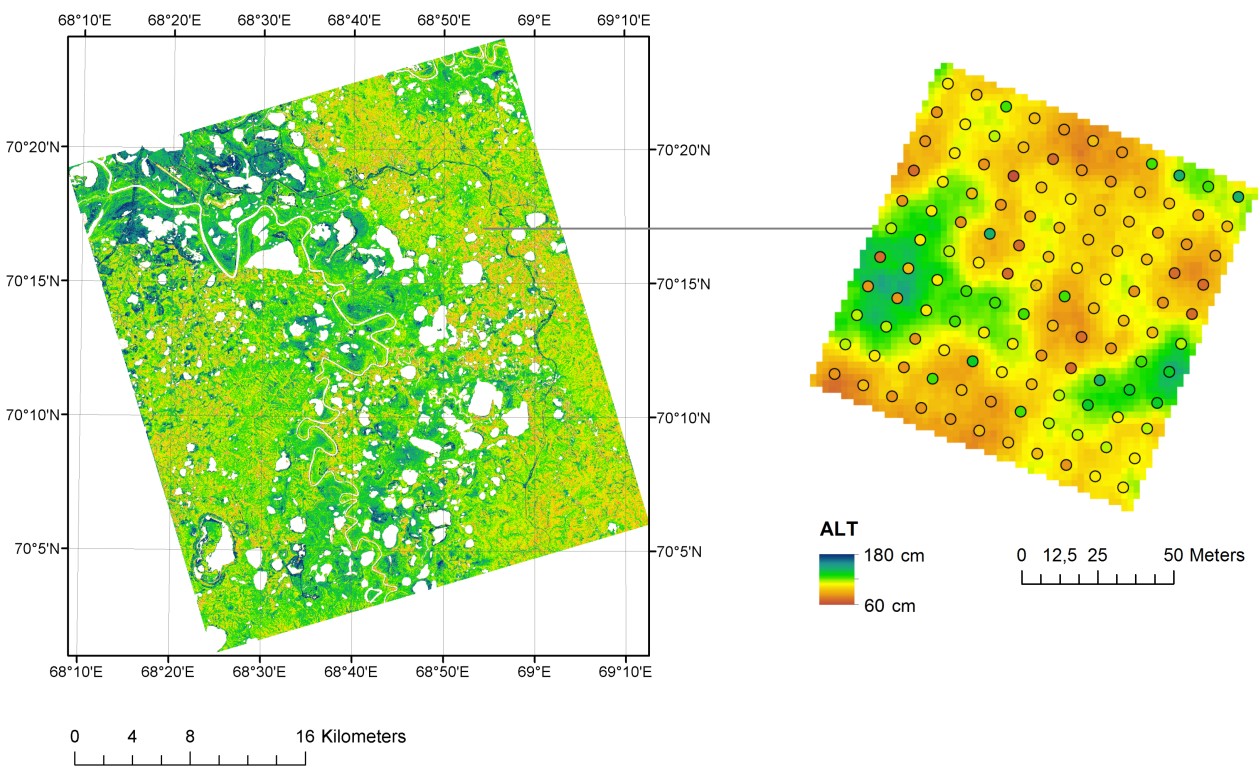

**Figure 7.** Left: Calculated ALT map for the entire TerraSAR-X scenes, based on TerraSAR-X backscatter values. Right: Calculated ALT map (background raster, based on X-band backscatter) for the CALM grid compared to in situ measurements (circles)

shading in the summer (Blok et al., 2010; Lawrence and Swenson, 2011; Pearson et al., 2013), leading to shallower ALT. In other publications (Domine et al., 2016; Myers-Smith and Hik, 2013) shrubs have been linked with ground warming due to the isolating effect of snow cover. Shrubs trap wind-blown snow and limit snow erosion by wind. It seems that the prevailing effect of cooling or warming may be caused by the length of winter or summer season. For the CALM site it can be shown that

shrubs coincide with deeper ALT (Figure 6). However it needs to be noted that shrubs are here found on saline clay, which, as stated before, can lead to higher ALT values measured by probing. In clayey saline soils (with ALT values > 130 cm at the CALM site) the freezing point is different from zero and probes can be inserted to greater depths (Leibman, 1998).

Soil moisture is another factor that influences microwave backscatter and ALT. It increases the backscatter amount due to the rise of the dielectric constant. It may also lead to higher ALT, caused by an increased heat flux and an enhanced conductive

transfer of heat into subsurface layers (Gross et al., 1990; Shiklomanov et al., 2010). For the Yamal region the measured moisture content is influenced strongly by lithology. While sandy soils are found to be dry, clayey saline soils often hamper infiltration and show higher soil moisture or even standing water. Although barren sandy areas are dryer, they have been shown to have maximal ALT (Leibman et al., 2015) due to higher heat flux, increased by infiltration of rain water. Cryptogam crust is typically found on high centred sandy polygons in the study area. For these points the following restrictions of the

used approach become apparent. Most points for which the modelled ALT shows high deviations from the in situ data have or lie close to spots of cryptogam crust. The volume scattering for areas with cryptogam crusts is expected to be very low (as described in the 'Results' section) but the recorded $\sigma_0$ is comparable to medium shrub heights which have a lower thaw depth. The higher $\sigma_0$ values could be due to higher surface roughness. The moisture content itself may also contribute to $\sigma_0$, however soil moisture values for areas with cryptogam crust have shown to be very low (see Figure 3). Heterogeneity regarding

vegetation coverage may also contribute. Especially sites with mixed types show high deviations (see Figure 6).

The redistribution of water and consequently also the occurring vegetation is controlled by lithology, but also by topography (see Figure 8), which has been shown useful in delineating ALT (Peddle and Franklin, 1993; Leverington and Duguay, 1996; Gangodagamage et al., 2014). Although we did not use topographic information in our study in order to derive ALT, it is indirectly introduced into our measurements through its effects on vegetation and moisture. For future studies the potential of

refining the backscatter approach by incorporating topographic information could be explored.

The comparison of the produced ALT map to the ALT measurements at the CALM grid showed that some points within the validation dataset from the CALM grid are not well represented by the map (Figure 7). These sites deviate also in the probe data. For instance for one of these points an ALT of > 170 cm was measured in 2014, which is about almost double the values of the surrounding points, and in 2015 only 116 cm were registered. Such an extreme variability of ALT could be explained by

an interface between a sandy active layer and a clayey permafrost at this location.

Regarding the delineated ALT map for the entire TerraSAR-X scene (Figure 7) it needs to be considered that neither the calibration data (VD sites), nor the validation site (CALM grid) included lowlands with extensive wetlands, as can be found in this region. Because of the lack of ALT measurements for these areas, the shown map is only representative for parts of the region, while lowlands would need further verification and could be masked out using a DEM. The higher ALT in drained lake

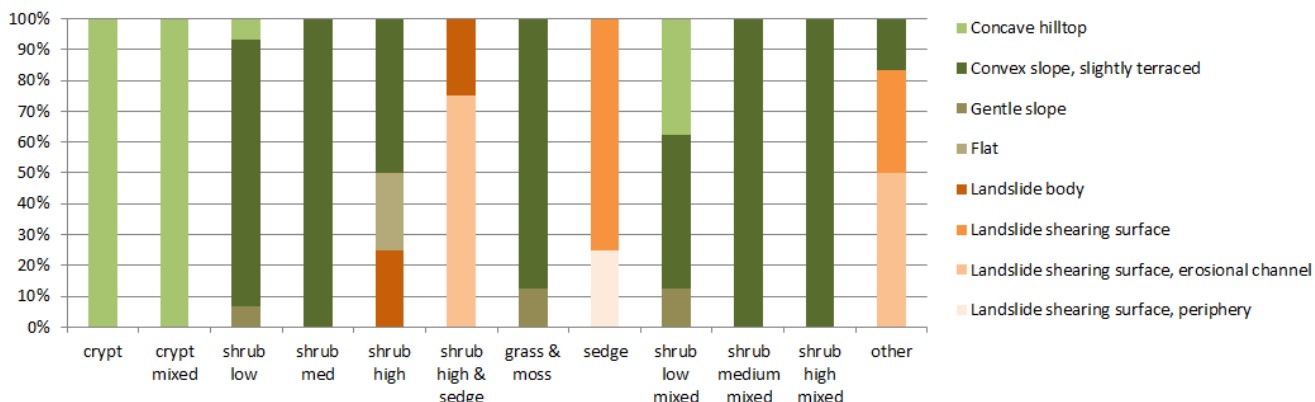

**Figure 8.** Comparison of topographic units (source CALM metadata, Earth Cryosphere Institute SB RAS) and vegetation type (survey of August 2015) at the CALM grid, illustrating the interrelation of certain topographic features and specific vegetation covers.

basins does agree with findings of Schaefer et al. (2015) around Barrow. They, however, argue that their algorithm is actually overestimating ALT at these locations.

The comparison of Landsat 8 derived NDVI values and ALT revealed a negative correlation (see Figure 5), which is contrary to most studies of other regions (Gangodagamage et al., 2014; McMichael et al., 1997). However Leibman et al. (2015) previously obtained similar results for the Yamal region at the CALM grid. Results suggest that NDVI is higher for sites with grasses, mosses and sedges than for medium to high shrubs (Salix) although the used near infrared is influenced by plant cell structure and vegetation biomass. These shrubs do, however, coincide with very wet soils (Figure 3) and are growing within partially waterlogged depressions. The sensitivity of NDVI to water (Raynolds and Walker, 2016) might have an impact over these sites. Furthermore a difference between the training datasets becomes apparent. Both, VD-1 and VD-2, lie within a similarly lower ALT range, but especially the acquisition of $22^{nd}$ July 2014 shows clearly higher NDVI values for VD-1 than for VD-2. This points to different vegetation types. A difference is also observed for the X-Band backscatter, where VD-1 shows slightly lower backscatter values than VD-2. The backscatter differences could be explained due to a slightly higher surface roughness, which is however not pronounced enough to have a sufficient influence on the site's snow cover or even ALT. Moreover vegetation at VD-2 is rather heterogeneous and also includes patches of sedges or reeds, which could increase the amount of backscatter. While VD-1 and to a small amount even VD-3 also shows sedge vegetation, these sandier sites show a different kind of grass (carex bigelowii instead of calamagrostis holmii). Moreover a clear difference between Landsat acquisitions of $22^{nd}$ July 2014 and $10^{th}$ August 2015 exists due to differences in vegetation and their respective phenological cycles (Bratsch et al., 2016), which are in turn influenced by terrain features and consequent draining conditions.

The advantages and disadvantages of the new approach with respect to previous studies could be demonstrated by application of the NDVI approach over the same site and using the same training data set. Investigations presented in this paper showed that the introduced approach of using TerraSAR-X backscatter values to delineate ALT values is suitable to provide higher than 30 m resolution estimates over unsaturated soils with ALT ranging from 40 cm to 140 cm in areas outside of high centred

sandy polygons. One previous study that was conducted within comparable ALT ranges used NDVI and land cover information respectively derived from Landsat data in combination with DEM data to derive three classes of ALT (Leverington and Duguay, 1996). A 93% agreement rate for three different ALT classes was obtained. In difference to our study Leverington and Duguay used "best-estimate" ALT values from either a pit value, or the average of pit and probe measurements, and fixed ALT classes.

In our study we used measurements for single points from grid points within a 100 x 100 m raster of high heterogeneity. All but one measured ALT value fall into only one ALT class (70 - 150 cm) used by Leverington and Duguay. The given accuracy can therefore not be compared.

The used TerraSAR-X data have been acquired in stripmap mode which has a swath width of 30 km. This sensor can however also acquire data over 270 km when using the Wide ScanSAR mode (40 m resolution). This would allow the transfer of the approach to larger regions. The so far investigated incidence angle range is however very limited. Differences in the described backscatter-ALT relationship due to incidence angle effects would need to be considered.

## 6  Conclusions

The common dependency of ALT and X-band backscatter values (HH, approximately 30 degree incidence angle) on land cover types as well as the interrelation of terrain, vegetation, soil moisture and snow cover yields a correlation which can be used to derive ALT with an RMSE of 20 cm over a long term monitoring site on central Yamal. It can be shown that in general higher ALT values correspond with higher backscatter values for the tested ALT range of 40 - 140 cm. The accuracy is lower over sites with mixed vegetation types within the pixels. Especially contributions from areas with little vegetation (cryptogam crusts) alter the relationship between backscatter and ALT. Soil moisture and/or surface roughness are influencing the signal over these sites. Polarimetric SAR analyses which allows to distinguish different scattering mechanisms might be suitable to tackle these issues. This could however not be tested due to unavailability of such satellite data. Results indicate a better performance than NDVI for higher ALT, but investigations with higher spatial resolution optical data would be required for confirmation.

*Author contributions.*  Barbara Widhalm has performed all data analyses, collected moisture and vegetation information at the CALM site and compiled the manuscript. Annett Bartsch has been contributing to the development of the concept of the approach as well as to the manuscript. Marina Leibmann and Artem Khomutov have collected the active layer measurements at the long term monitoring sites as well as the vegetation information at sites outside of the ALT monitoring sites and contributed to the interpretation and discussion of the results.

*Acknowledgements.*  This work was supported by the Austrian Science Fund under Grant [I 1401] and Russian Foundation for Basic Research Grant 13-05-91001-ANF-a (Joint Russian–Austrian project COLD-Yamal). TerraSAR-X data have been available by DLR through PI agreement LAN1706 and HYD2522, and Tandem-X data through HYDR0226.

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
