# Peer review of "Active Layer Thickness Estimation from X-Band SAR Backscatter Intensity"

_The Cryosphere, 2016_

## Referee Comment (RC1) · Anonymous Referee #1 · 20 Sep 2016

Widhalm et at. [2016] relate x-band backscatter to in situ measurements of Active Layer Thickness (ALT). They found the backscatter correlates well with ALT for ALT greater than about 70 cm, but not so much for smaller values of ALT. Essentially, the X-band backscatter measures the vegetation characteristics that control or influence ALT, such as vegetation height. The idea is new, interesting, and shows potential for mapping ALT over larger areas. However, the paper will require major revisions before it is ready for publication. I have several major comments: 1) The authors need to incorporate more physical interpretations of volumetric scattering and vegetation characteristics as they relate to ALT. The authors emphasize the technique, but if they want to capture the imagination of this journal's readership, they must expand the physical interpretation. We know vegetation influences ALT based on probing data and plant type, but these backscatter measurements offer direct, pixel-by-pixel measurements of the vegetation

physical characteristics associated with ALT variability. How does this effort relate to efforts to use backscatter to measure biomass? What physical characteristics drive the backscatter strength: vegetation height, total biomass, or leaf density? How do these characteristics relate to the thermodynamics that drive ALT? This is the true science advancement of this paper. 2) The authors must include the exact mathematical form of the curve fits, the regression coefficients, and associated uncertainty. Others cannot reproduce these results without the form and coefficients. The uncertainty is absolutely essential to interpreting the results. 3) The authors should include a map of ALT and uncertainty for the entire X-band scene, not just the CALM sites. This is what the readers and I really want to see. Such maps are more useful and interesting to the readers than the curve fit plots. I recognize the technique does not work as well for ALT < 70 cm, but this is fine as long as the uncertainty reflects this. 4) Drop all functions except the linear curve fit. Figures 5 and 6 clearly indicate no statistical difference between the three functions within the data range. Using three curve fits adds volume, but not value to the paper. I know of no theoretical basis for choosing any one of these functions over the others, so I recommend the authors stick with the simplest form: the linear curve fit. 5) Drop the comparison to NDVI. The backscatter technique does appear more robust than the NDVI technique, but I am not sure how much value this adds to the paper. The NDVI method is only one of several methods identified by the authors, and definitely not the best, so I do not see why the authors focus on this particular technique. Besides, the stated goal was to evaluate backscatter, not compare with the NDVI method. 6) Rewrite the manuscript using the active voice. Readers find the passive voice difficult to follow. Minor Comments: P1, L3, Table 1; P2, L16; P2, L23-4: The techniques listed in Table 1 are not 'limited' to ALT < 70 cm. The wording implies the techniques do not work for deeper active layers, which is not true. For example, Schaefer et al. [2015] and Gangodagamage et al. [2014] estimated similar ALT because that is the actual ALT around Barrow, AK. Other studies using similar techniques measured ALT > 70 cm in different areas, such as Liu et al. [2012] around Prudhoe Bay and Pastick et al. [2013] around Yukon Flats. P1, L16: The authors need

more recent and comprehensive references for carbon emissions, such as Schaefer et al., [2011] or Schuur et al. [2013]. P1, L4-5: Although this may be true for Yamal, it is not true in general. All the papers in Table 1 as well as others not included show spatial patterns of ALT. P2, L6-26: The authors need to rewrite this section with a more logical flow of ideas. The authors need to clearly separate modeling techniques to estimate ALT from the upscaling techniques based on probing. They start out with the Stefan model and Kudryavtsev models, but suddenly switch to Gangodagamage et al. [2014], who used a probing data to scale up lidar data, a technique that is remarkably similar to what the authors describe with backscatter. Then they switch to InSAR, then to soil moisture, and back to upscaling probing data. P2, L31-4: The authors need to explain why they think backscatter relates to ALT. The authors state their hypothesis, but do not explain why backscatter should relate to ALT. What have people done in the past relative to backscatter and biomass in permafrost regions? How does biomass relate to ALT? How is x-band an advantage? Why not use C-band or L-band? P3, Fig1: This figure should outline the X-band scenes used in the study. P4, Tab1: The authors should identify the general area of the studies, such as Barrow for Gangodagamage et al. [2014]. The text should explain that all these studies except Schaefer et al. [2015] used regressions of ALT against a remotely sensed characteristic to scale up probing data. The exception is Schaefer et al. [2015], who used InSAR to measure seasonal subsidence and did not use probing regressions. P6, L5; P8, Fig2: The authors need to explain this figure, which I did not understand. One sentence is not enough. How did the authors use this data? P6, L26-31: The authors should delete this paragraph. TanDEM-X data can be used for InSAR, but that is not what the authors did. The authors essentially created regressions of ALT vs backscatter, not InSAR. P7, L23-4: Why would a relationship between NDVI and ALT indicate a relationship between backscatter and ALT? P7, L27 to P8, L2: The authors need to explain the technical terms associated with SAR that the broader audience of the Cryosphere will not understand. They need to define: Range Doppler correction, radiometric normalization, sigma-zero (the primary parameter), speckle, and near neighborhood. The average

permafrost scientists will have no clue what these terms mean. P8, L6: What ALT values? From the CALM sites? Later the authors use the same ALT to 'validate' the regression, which seems circular. P8, L4-5: The authors need to explain how and why they choose these classes. P8, L7-8: Stick to the linear function, as I state above. Show the exact form here. P8, L20: The authors need to explain figures 3, 4, and 5. One sentence each does not suffice. P9, L7-8: The authors must explicitly define 'coefficient of determination in the methods section, exactly how they calculated it, and what it means. P8, L8: The authors need to estimate the ALT uncertainty as a function of backscatter. RMSE is OK, but we really need an uncertainty estimate. P11, L7-8: The authors need to explain why September would differ from August in terms of backscatter. I strongly suspect that leaves have fallen off the plants and the ground surface has started to freeze, altering the backscattering characteristics. P11, L14-5: Please explain the 'restrictions of the used approach.' P11, L19-20: The moisture content will definitely contribute to backscatter, but the authors need to explain how. The authors should identify the expected penetration depths for dry and wet tundra. P 13, L9-11: The backscatter technique performs better than the NDVI technique, but this does not support or refute the initial hypothesis that you can use backscatter to estimate ALT. I suggest deleting this. P13, L12-4: I agree that you can scale this technique to larger areas and suggest you add a map of ALT for the entire x-band scene. This is what the readers really want to see.
* * *

---

## Referee Comment (RC2) · Anonymous Referee #2 · 21 Sep 2016

This paper explores the relationship between X-band SAR backscatter and active layer thickness, on the premise that the amount of vegetation exerts control over both the thawing depth of the ground and SAR backscatter values; the SAR backscatter values being driven by volume scattering which is controlled by the height and structure of the vegetation. The relationship has some merit and the authors demonstrate a reasonable relationship over areas with ALT thicker than 70 cm. In areas with ALT less than this there are other land-cover types that produce similar backscatter values to the shrub cover, making the relationship unreliable.

I found the paper to be quite thorough and well presented. While the results are not a 'silver bullet' answer to SAR remote sensing of ALT, they make a contribution to the layers of science from which more complex, but more reliable, models might evolve.

[Figure]

I have only minor suggestions for the authors, the English is generally good but should be reviewed in a few places, as included in the points below:

Pg 2: Line 11 'a great potential' -> 'great potential' Pg 5:10 'is resulting' -> 'results' Pg 6:23: stating the strip length is confusing since it is not the same as the full scene size that is actually analysed. Pg 6:24 maybe say how many scenes rather than just 'all'. Pg 7:8-9 'to the in this study' clarify Pg 7: 12-19. I would have thought that vegetation water content and hence dielectric properties would also influence backscatter values, but this is not mentioned at all. Only vegetation structure is mentioned. An expanded discussion of the interaction of vegetation and backscatter would be good. Pg 7:23 'in the surrounding.' Do you mean 'in the surrounding area'? Pg 7:30 It was not clear to me if the radiometric normalization applied was to normalize radiometry between images, or to remove terrain related (incidence angle) radiometric effects within images. Clarify. Pg 8:1 suggest replacing 'account' with 'subdue' Pg 8:19 replace 'like' with 'as' Pg 13:9 'can be however not used' -> 'cannot however, be used'.

Figure 1. The location map annotation needs to be enlarged. It was not readable. Figure 2. 'Flat slope' seems a strange and contradictory name

―――――――――――――――――――

---

## Referee Comment (RC3) · Anonymous Referee #1 · 23 Sep 2016

Although all three reviewers used different words, I think we all say essentially the same thing: the authors need to create a link between those vegetation characteristics (or land cover classes) that drive both x-band backscatter and ALT dynamics. Overlapping vegetation characteristics that influence both backscatter and ALT would simultaneously provide both a motivation for the study and an explanation of the results. The authors find that backscatter correlates with ALT, at least for ALT > 70 cm, clearly indicating a relationship. The missing piece of the manuscript and the significant scientific advance is explicitly identifying what vegetation characteristics drive this relationship. Reviewer 1

---

## Short Comment (SC1) · 23 Sep 2016

Active layer thickness (ALT) is a fundamental variable in permafrost studies. I believe that this study is a useful contribution to the body of literature on ALT mapping from aerial/satellite data. However, the manuscript seems to require some conceptual-logical reworking. This seems not to have been highlighted by the two referees that posted their reviews prior to this short comment. Please consider the following.

For example, the abstract states: "This study shows that the mutual dependency of ALT and TerraSAR-X backscatter on land cover types induces a connection of both parameters." This reads as a conclusion, but arguably is the rationale for your study. That active layer thickness depends on land cover type, and that backscatter intensity varies with land cover, is well known. The key question thus may be related to: are

spatial variations in x-band backscatter intensity related to land cover differences use-ful for estimating active layer thickness? Better: are land cover classes derived from x-band backscatter intensity useful for estimating active layer thickness? In this sense, the title and the conceptual-logical structure of the manuscript may not be adequate. To finalize my comment, consider the first sentences of the Introduction's last para-graph: "SAR backscatter intensity has so far not been investigated for ALT estimation. Radar backscatter at X-band is also related to vegetation coverage, especially shrubs (Duguay et al., 2015, similarly to the NDVI)." The second sentence implies that x-band backscatter depends not only on land cover but also, directly, on ALT. This is at odds with the methodology, etc.

---

## Author Comment (AC1) · 28 Sep 2016

We thank the referee for the thorough review and the many useful comments, which we will incorporate into the revised manuscript and reply to in detail during the final response phase. For the ongoing interactive discussion we would like to respond to some comments.

Response to comment 1): We thank you for pointing out the lack of information regarding vegetation characteristics that drive both X-band backscatter and ALT dynamics. We will revise the paper accordingly.

Response to comment 5): We incorporated the NDVI technique in order to compare the X-band backscatter method to an already established "traditional" method. Furthermore the NDVI was used in order to discuss the differences between the sites VD-1

and VD-2, which show similar ALT values while backscatter values seem to differ. We appreciate your comment and will revise our manuscript in order to clarify our intentions.

Response to comment on P1,L3, Table 1; P2,L16; P2,L23-4: '...Other studies using similar techniques measured ALT > 70 cm in different areas, such as Liu et al. [2012] around Prudhoe Bay and Pastick et al. [2013] around Yukon Flats.' We thank you for pointing out these papers.

Response to comment on P6,L5; P8, Fig2: 'The authors need to explain this figure, which I did not understand. One sentence is not enough. How did the authors use this data?' The figure shows the relationship of topography and vegetation types and that certain topographic features are linked to specific vegetation covers. As is stated in the introduction, both slope and vegetation cover can be linked to ALT and as can be seen from the figure, they are also connected to each other. We will clarify this and although we did not use any topographic information in our delineation of ALT we will point out the potential of refining the backscatter approach by incorporating topographic information in the revised manuscript.

Response to comment on P6, L26-31: 'The authors should delete this paragraph. TanDEM-X data can be used for InSAR, but that is not what the authors did. The authors essentially created regressions of ALT vs backscatter, not InSAR.' The DEM was used in our study to perform the Range-Doppler Terrain Correction. The paragraph describes its origin. We suggest renaming the chapter from 'TerraSAR-X acquisitions' to 'TerraSAR-X data' in order to prevent confusions.

Response to comment on P8, L6: 'What ALT values? From the CALM sites? Later the authors use the same ALT to 'validate' the regression, which seems circular.' The CALM site data were only used for validation and the map over the entire site (not only the in situ points) produced in order to demonstrate and discuss the uncertainties. Sites VD-1, VD-2 and VD-3 were used for calibration only. We will revise the manuscript to

emphasize this.

---

## Author Comment (AC2) · 28 Sep 2016

We thank the referee for the review of our manuscript and appreciate the comments. We will incorporate them and address them in detail during the final response phase. For now we would like to respond to some comments.

Response to comment on Pg 7: 12-19. *'I would have thought that vegetation water content and hence dielectric properties would also influence backscatter values, but this is not mentioned at all. Only vegetation structure is mentioned. An expanded discussion of the interaction of vegetation and backscatter would be good.'*
Thank you for your comment. This was also pointed out to us in other comments. We will expand the manuscript accordingly in the revised version.

[Figure]

Response to comment on Pg 7:30 *'It was not clear to me if the radiometric normaliza-tion applied was to normalize radiometry between images, or to remove terrain related (incidence angle) radiometric effects within images. Clarify.'*
The radiometric normalization was applied to remove terrain related effects. We will revise the manuscript to clarify this.

Response to comment on Figure 1. *'The location map annotation needs to be enlarged. It was not readable.'*
We assume this comment refers to the annotations in the bottom right corner of the left location map. As it was not possible to enlarge it we put it in the figure description.

Response to comment on Figure 2. *"Flat slope' seems a strange and contradictory name.'*
We thank you for pointing this out. The names of this figure describing the topography of the CALM grid correspond to the metadata found for this CALM grid. Here three types of slope surfaces were subdivided: concave, flat and convex. We will rename the class from 'flat slope' to 'flat' to prevent confusions.

---

## Author Response (AR1)

**REFEREE #1:**

**I have several major comments:**

1) The authors need to incorporate more physical interpretations of volumetric scattering and vegetation characteristics as they relate to ALT. The authors emphasize the technique, but if they want to capture the imagination of this journal's readership, they must expand the physical interpretation. We know vegetation influences ALT based on probing data and plant type, but these backscatter measurements offer direct, pixel-by-pixel measurements of the vegetation physical characteristics associated with ALT variability. How does this effort relate to efforts to use backscatter to measure biomass? What physical characteristics drive the backscatter strength: vegetation height, total biomass, or leaf density? How do these characteristics relate to the thermodynamics that drive ALT? This is the true science advancement of this paper.

We thank you for pointing out the lack of information regarding vegetation characteristics that drive both X-band backscatter and ALT dynamics. We have revised the paper and expanded the discussion section accordingly. We would like to refer to the file which includes the tracked changes.

**2) The authors must include the exact mathematical form of the curve fits, the regression coefficients, and associated uncertainty. Others cannot reproduce these results without the form and coefficients. The uncertainty is absolutely essential to interpreting the results.**

We now included the form and std error for the regression lines. Functions are now also included in the figures

The mathematical form of the regression line is given in Figure 6. The standard error of the intercept is 14.00 and 0.96 for the slope respectively.

The standard error of the intercept is 9.21 for 22nd July 2014 and 11.60 for 10th August 2015 and 15.80 and 20.21 for the slope respectively.

**3) The authors should include a map of ALT and uncertainty for the entire X-band scene, not just the CALM sites. This is what the readers and I really want to see. Such maps are more useful and interesting to the readers than the curve fit plots. I recognize the technique does not work as well for ALT < 70 cm, but this is fine as long as the uncertainty reflects this.**

We added the ALT map for the entire X-band scene. We included the std error of slope and intercept of the regression line. Uncertainties have been published for the probe measurements from the CALM site (4 cm) and we therefore included  $\chi^2$  for the CALM grid (which is validation site only). We also added R2 for every VD site separately to illustrate the accuracy of the product (see Table 2). We further removed the statement that the approach is limited to ALT depths larger than 70 cm as this cannot be sufficiently quantified over our study area due to lack of ALT sampling data (and limited occurrence at the site) within the lower range.

**4) Drop all functions except the linear curve fit. Figures 5 and 6 clearly indicate no statistical difference between the three functions within the data range. Using three curve fits adds volume, but not value to the paper. I know of no theoretical basis for**

**choosing any one of these functions over the others, so I recommend the authors stick with the simplest form: the linear curve fit.**

We revised the paper and dropped the other functions. (See file of tracked changes + adaption of Figures and Table 2)

**5) Drop the comparison to NDVI. The backscatter technique does appear more robust than the NDVI technique, but I am not sure how much value this adds to the paper. The NDVI method is only one of several methods identified by the authors, and definitely not the best, so I do not see why the authors focus on this particular technique. Besides, the stated goal was to evaluate backscatter, not compare with the NDVI method.**

We incorporated the NDVI technique in order to compare the X-band backscatter method to an already established "traditional" method. Furthermore the NDVI was used in order to discuss the differences between the sites VD-1 and VD-2, which show similar ALT values while backscatter values seem to differ. We appreciate your comment and revised our manuscript in order to clarify our intentions.

...Its thickness is typically measured locally, but a range of methods, which utilize information from satellite data, exist. Mostly, the Normalized Difference Vegetation Index (NDVI) obtained from optical satellite data is used as proxy.

... We therefore investigated the relationship between ALT and X-Band SAR backscatter of TerraSAR-X (averages for 10 x 10 m window) in order to explore the possibility of delineating ALT on a continuous and larger spatial coverage in this area and compare it to the already established method of using NDVI from Landsat (30 m).

**6) Rewrite the manuscript using the active voice. Readers find the passive voice difficult to follow.**

**We adapted the manuscript in places, where we think the passive voice could lead to confusions.**

We therefore investigated the relationship between ALT and X-Band SAR backscatter of TerraSAR-X ...

We used in situ records from several sites in the proximity of a long-term monitoring site on Yamal and discuss the results with respect to previous approaches which use remotely sensed information as proxy for ALT.

In August 2015 we carried out a dedicated vegetation survey of each CALM grid point, where we determined the dominant vegetation cover within a 3 x 3 m area.

Furthermore we conducted moisture measurements at the CALM grid. We used the Delta-T Wet Sensor with HH2 handheld to measure the moisture content of the top 5 cm at each grid point on three dates in August 2015.

We compiled a data set based on TerraSAR-X which allows the investigation of this relationship.

**Minor Comments:**

 P1, L3, Table 1; P2, L16; P2, L23-4: The techniques listed in Table 1 are not 'limited' to ALT < 70 cm. The wording implies the techniques do not work for deeper active layers, which is not true. For example, Schaefer et al. [2015] and Gangodagamage et al. [2014] estimated similar ALT because that is the actual ALT around Barrow, AK. Other studies using similar techniques measured ALT

**> 70 cm in different areas, such as Liu et al. [2012] around Prudhoe Bay and Pastick et al. [2013] around Yukon Flats.**

We removed the statement that these techniques were limited to ALT < 70 cm and clarify that they were simply conducted in areas with shallow ALT. We thank you for pointing out these papers, however also Liu et al. [2012] found ALT of 50 – 80 cm which is in the lower range of values we are dealing with in our study. Pastick et al. [2013] on the other hand not only used remote sensing data, but also climatic data. It is therefore not a pure remote sensing approach.

The applicability has been demonstrated mostly for shallow depths of Active Layer Thickness (ALT) below approximately 70 cm.

Most previous remote sensing approaches (Leverington and Duguay, 1996; McMichael et al., 1997; Sazonova and Romanovsky, 2003; Schaefer et al., 2015) have utilized data with spatial resolutions of 30 m and coarser and were conducted in areas with shallow ALT (less than approximately 70 cm, see Table 1).

**• P1, L16: The authors need more recent and comprehensive references for carbon emissions, such as Schaefer et al., [2011] or Schuur et al. [2013].**

Thank you for suggesting these papers; we included them as references

At global scale, increased ground temperatures could facilitate further climatic changes by releasing greenhouse gases that are currently sequestered in 20 the upper layer of permafrost by increasing the annual thaw depth (Kane et al., 1991; Gomersall and Hinkel, 2001; Shiklomanov and Nelson, 1999; Schaefer et al., 2011; Schuur et al., 2015).

**• P1, L4-5: Although this may be true for Yamal, it is not true in general. All the papers in Table 1 as well as others not included show spatial patterns of ALT.**

The statement referred to the sentence before about Yamal. We have now merged them.

Changes in active layer thickness have been already observed for Yamal (Leibman et al., 2015), where active layer thickness spatial patterns are unknown outside of the sites with in situ measurements.

• P2, L6-26: The authors need to rewrite this section with a more logical flow of ideas. The authors need to clearly separate modeling techniques to estimate ALT from the upscaling techniques based on probing. They start out with the Stefan model and Kudryavtsev models, but suddenly switch to Gangodagamage et al. [2014], who used a probing data to scale up lidar data, a technique that is remarkably similar to what the authors describe with backscatter. Then they switch to InSAR, then to soil moisture, and back to upscaling probing data.

We appreciate your comment and hope that the made changes provide a better flow. We would like to refer to the file which includes the tracked changes.

• P2, L31-4: The authors need to explain why they think backscatter relates to ALT. The authors state their hypothesis, but do not explain why backscatter should relate to ALT. What have people done in the past relative to backscatter

**and biomass in permafrost regions? How does biomass relate to ALT? How is x-band an advantage? Why not use C-band or L-band?**

Very little work has been done on shrub biomass in tundra with X-band. Only Duguay et al. 2015 looked into shrub height. But we have now added some general comments on volume scattering. See the last paragraph of the introduction section of the revised manuscript.

• P3, Fig1: This figure should outline the X-band scenes used in the study.

We added the outline to the map

• P4, Tab1: The authors should identify the general area of the studies, such as Barrow for Gangodagamage et al. [2014]. The text should explain that all these studies except Schaefer et al. [2015] used regressions of ALT against a remotely sensed characteristic to scale up probing data. The exception is Schaefer et al. [2015], who used InSAR to measure seasonal subsidence and did not use probing regressions.

We added a column stating the study areas (see file of tracked changes) and revised the mentioned text within the introduction chapter.

Recently, subsidence rates have been used as input for modelling ALT (Schaefer et al., 2015). Synthetic Aperture Radar has been exploited using interferometric analyses (InSAR, provides seasonal ground subsidence) in combination with soil properties to estimate ALT without using empirical relationships with probing data (Schaefer et al., 2015).

**• *P6, L5; P8, Fig2: The authors need to explain this figure, which I did not understand. One sentence is not enough. How did the authors use this data?**

The figure shows the relationship of topography and vegetation types and that certain topographic features are linked to specific vegetation covers. As is stated in the introduction, both slope and vegetation cover can be linked to ALT and as can be seen from the figure, they are also connected to each other. Further elaboration can be found in the discussion section of the revised manuscript.

Although we did not use any topographic information in our delineation of ALT we point out the potential of refining the backscatter approach by incorporating topographic information in the revised manuscript.

Figure 2. Comparison of topographic units (source CALM metadata, Earth Cryosphere Institute SB RAS) and vegetation type (survey of August 2015) at the CALM grid, illustrating the interrelation of certain topographic features and specific vegetation covers.

The redistribution of water and consequently also the occurring vegetation is in turn controlled by lithology, but also by topography (see Figure 2), which has been shown useful in delineating ALT (Peddle and Franklin, 1993; Leverington and Duguay, 1996; Gangodagamage et al., 2014). Although we did not use topographic information in our study in order to derive ALT, it is indirectly introduced into our measurements through its effects on vegetation and moisture. For future studies the potential of refining the backscatter approach by incorporating topographic information could be explored. • P6, L26-31: The authors should delete this paragraph. TanDEM-X data can be used for InSAR, but that is not what the authors did. The authors essentially created regressions of ALT vs backscatter, not InSAR.

The DEM was used in our study to perform the Range-Doppler Terrain Correction. The paragraph describes its origin. We renamed the chapter from 'TerraSAR-X acquisitions' to 'X-band data' in order to prevent confusions.

**• P7, L23- 4: Why would a relationship between NDVI and ALT indicate a relationship between backscatter and ALT?**

We made the following changes in the manuscript to clarify this:

Based on results of previous studies which utilized the vegetation index NDVI (Table 1) and the mutual dependency of NDVI and radar backscatter on vegetation coverage, it is expected that a relationship between X-Band backscatter measurements and ALT is given.

• P7, L27 to P8, L2: The authors need to explain the technical terms associated with SAR that the broader audience of the Cryosphere will not understand. They need to define: Range Doppler correction, radiometric normalization, sigma-zero (the primary parameter), speckle, and near neighborhood. The average permafrost scientists will have no clue what these terms mean.

**We added short descriptions of the used terms.**

Utilising the software NEST (Next ESA SAR Toolbox) Range-Doppler Terrain Correction was performed with a TanDEM-X Intermediate DEM (~ 12 m resolution) in order to orthorectify the images and to compensate for distortions due to topographical variations and the tilt of the sensor. Images were processed to a pixel spacing of 2 m and a radiometric normalization was applied to account for incidence angle dependent sensitivity. The so called resulting  $\sigma_0$  values were then converted into dB. The term backscatter refers in the following to these values, which represent the normalised measure of the radar return. SAR data are affected by so called speckle which is a noiselike effect. It can be understood as an interference phenomenon due to a number of scatterers within each resolution cell. The images were therefore further averaged over time and a spatial filter (average value of the cells in the neighbourhood was calculated: 5 x 5 cells – 10 m) was applied to subdue this noise.

**• *P8, L6: What ALT values? From the CALM sites? Later the authors use the same ALT to 'validate' the regression, which seems circular.**

The CALM site data were only used for validation. The map over the entire CALM site (not only the in situ points) was produced in order to demonstrate and discuss the uncertainties. Sites VD-1, VD-2 and VD-3 were used for calibration only. We revised the manuscript to emphasize this.

Backscatter as well as NDVI values for the sites VD-1, VD-2 and VD-3 were extracted and compared to the mean ALT values of 2014 and 2015 of each measuring point.

Validation has been undertaken using ALT measurements at the CALM grid. RMSE was calculated representing the modelled ALT (linear regression of all VD sites) versus the measured ALT at the CALM site.

• *P8, L4-5: The authors need to explain how and why they choose these classes.*

The chosen class range allows a representative number of values for the statistics. We now clarified this in 'Methodology' section.

ALT values were separated into 10 cm classes and the backscatter classes ranged 1 dB, allowing a representative number of sampling points per class (7 and 13 points minimum per class respectively).

**• *P8, L7-8: Stick to the linear function, as I state above. Show the exact form here.**

We now included the forms (see above).

• P8, L20: The authors need to explain figures 3, 4, and 5. One sentence each does not suffice.

We extended the text explaining figures 3, 4, and 5 and want to point out that they are further discussed within the Discussion chapter.

The assumption that backscatter increases with increasing amount of vegetation could be confirmed for the Vaskiny Dachi area (Figure 2). There is a difference of about 2 dB between the median for shrubs less than 20 cm and those larger than 60 cm. σ0 values for the grass/sedge class do however exceed these values. Cryptogam crust backscatter is at the same order of magnitude as shrubs between 20 and 60 cm height. Although the sparse vegetation at these areas consists mostly of lichen and volume scattering within is negligible, these spots often show higher surface roughness or even hummocks which may lead to a rise in backscatter amount. The boxplot of Figure 3 which shows the relationship between soil moisture and vegetation types at the CALM grid reveals that lowest soil moisture is encountered for areas with cryptogam crust, while higher shrubs and especially sedges dominate in areas of high soil moisture.

Class statistics (Figure 4) indicate a relationship between  $\sigma$ 0 and larger thaw depths (> 70 cm). Low backscatter values dominate in areas with low ALT and high backscatter values coincide with high ALT. However, the median  $\sigma$ 0 for shallow ALT does not decrease with decreasing ALT at the same rate as for deeper ALT.

The Scatterplots of the filtered  $\sigma$ 0 and ALT values of the sites VD-1 to VD-3 also indicate this correlation (see Figure 5). It also becomes apparent that the change in slope, that is visible in the boxplot of Figure 4, is caused by different backscatter values of the sites VD-1 and VD-2 (Figure 5). Nevertheless  $\sigma$ 0 increases generally with increasing ALT.

Figure 4. Boxplots showing median, minimum and maximum values, first and third quartile and outliers of the ALT and backscatter values of the sites VD-1, VD-2 and VD-3. Left: σ0 statistics for Active Layer Thickness (ALT) classes (10 cm). Right: ALT statistics for backscatter classes (1 dB).

**• P9, L7-8: The authors must explicitly define 'coefficient of determination in the methods section, exactly how they calculated it, and what it means.**

We included the formula and definition.

$$R^2 = 1 - \frac{\sum (y_i - f_i)^2}{\sum (y_i - \bar{y})^2}$$

The coefficient of determination (  $\sum^{(y_i - \bar{y})^2}$ , the proportion of the variance in the dependent variable that is predictable from the independent variable) was calculated for ...

• *P8, L8: The authors need to estimate the ALT uncertainty as a function of backscatter. RMSE is OK, but we really need an uncertainty estimate.*

According to the suggestion above we included the form of the regression and the std error as well as  $\chi^2$ . Furthermore we calculated R2 for every VD site separately to illustrate the accuracy of the product.

• P11, L7-8: The authors need to explain why September would differ from August in terms of backscatter. I strongly suspect that leaves have fallen off the plants and the ground surface has started to freeze, altering the backscattering characteristics.

We now included the explanation, which you already indicated. Although freezing would also alter the backscatter a weather station near the respective research area doesn't indicate freezing in the respective year.

A season related reduction of leaves on shrubs could lead to a decrease in volume scattering and therefore lower backscatter values.

• P11, L14-5: Please explain the 'restrictions of the used approach.'

The explanation is given in the next sentences. We have now changed the sentence to clarify this.

For these points the following restrictions of the used approach become apparent.

**• P11, L19-20: The moisture content will definitely contribute to backscatter, but the authors need to explain how. The authors should identify the expected penetration depths for dry and wet tundra.**

We included an explanation for the influence of moisture content. However we removed the remark on penetration depth at P11 as it didn't offer any solution for the discussed problem. Furthermore we are not aware of any available studies on penetration depth of X-band SAR for dry or wet tundra. But Zwieback et al. (2016,

http://ieeexplore.ieee.org/document/7486090/) observed that wetting and drying lead to phase closure errors at X-band over tundra, indicating that penetration depth is indeed changing with moisture content.

The overall backscatter intensity of a certain surface area is also influenced by surface roughness with respect to the wavelength (3.1 cm), as well as by soil moisture. A higher moisture content leads to a higher dielectric constant and therefore higher backscatter values and additionally a reduction in penetration depth.

Soil moisture is another factor that influences radar backscatter as well as ALT. It increases the backscatter amount due to the rise of the dielectric constant and may lead to higher ALT, caused by an increased heat flux and an enhanced conductive transfer of heat into subsurface layers (Gross et al., 1990; Shiklomanov et al., 2010). For the Yamal region the measured moisture content is influenced strongly by lithology. While sandy soils are found to be dry, clayey saline soils often hamper infiltration and show higher soil moisture or even standing water. Although barren sandy areas are dryer, they have been shown to have maximal ALT (Leibman et al., 2015) due to higher heat flux, increased by infiltration of rain water.

• *P* 13, L9-11: The backscatter technique performs better than the NDVI technique, but this does not support or refute the initial hypothesis that you can use backscatter to estimate ALT. I suggest deleting this.

We now deleted this paragraph in the revised manuscript.

• P13, L12-4: I agree that you can scale this technique to larger areas and suggest you add a map of ALT for the entire x-band scene. This is what the readers really want to see.

We now added the map.

**REFEREE #2**

**• Pg 2: Line 11 'a great potential' -> 'great potential'**

Thank you for your comments pointing out where to improve our English. Changes have been made accordingly.

- Pg 5:10 'is resulting' -> 'results'
  - has been adapted
- Pg 6:23: stating the strip length is confusing since it is not the same as the full scene size that is actually analysed.

We now removed this statement.

• Pg 6:24 maybe say how many scenes rather than just 'all'.

The number of scenes has now been included.

**• Pg 7:8-9 'to the in this study' clarify**

The text has been expended accordingly

**• Pg 7: 12-19. I would have thought that vegetation water content and hence dielectric properties would also influence backscatter values, but this is not mentioned at all. Only vegetation structure is mentioned. An expanded discussion of the interaction of vegetation and backscatter would be good.**

Thank you for pointing this out. We expanded the discussion on this topic and would like to refer especially to the changes within the Introduction chapter (see file tracking changes).

• Pg 7:23 'in the surrounding.' Do you mean 'in the surrounding area'?

Yes; this has been adapted.

• Pg 7:30 It was not clear to me if the radiometric normalization applied was to normalize radiometry between images, or to remove terrain related (incidence angle) radiometric effects within images. Clarify.

Thank you for pointing this out. We revised the manuscript and clarified this:

Images were processed to a pixel spacing of 2 m and a radiometric normalization was applied to account for incidence angle dependent sensitivity.

- Pg 8:1 suggest replacing 'account' with 'subdue'
  - has been adapted
- Pg 8:19 replace 'like' with 'as'

- has been adapted
- Pg 13:9 'can be however not used' -> 'cannot however, be used'.
  - has been adapted

**• Figure 1. The location map annotation needs to be enlarged. It was not readable.**

We assume this comment refers to the annotations in the bottom right corner of the left location map. As it was not possible to enlarge it we put it in the figure description.

**• Figure 2. 'Flat slope' seems a strange and contradictory name**

We thank you for pointing this out. The names of this figure describing the topography of the CALM grid correspond to the metadata found for this CALM grid. Here three types of slope surfaces were subdivided: concave, flat and convex. We renamed the class from 'flat slope' to 'flat' to prevent confusions.

**Active Layer Thickness Estimation from X-Band SAR Backscatter Intensity**

Barbara Widhalm1, Annett Bartsch1,2, Marina Leibman3,4, and Artem Khomutov3,4

1Zentralanstalt für Meteorologie und Geodynamik, 1190 Vienna, Austria

2Vienna University of Technology, 1040 Vienna, Austria

[revised manuscript text omitted]

8 cm)                                                                                                                                                       |

---

## Author Response (AR2)

Dear Andreas Kääb,

Thank you for your comments. We included the suggested final changes to the manuscript (see below). Concerning your comment in tc-2016-177-comments-to-author.pdf on page 18 line 2 we would like to refer to the explanation given on page 6 line 3.

[revised manuscript text omitted]